# Antithrombotic Strategies in Patients with Atrial Fibrillation and Acute Coronary Syndromes Undergoing Percutaneous Coronary Intervention

**DOI:** 10.3390/jcm11030512

**Published:** 2022-01-20

**Authors:** Leonardo De Luca, Raffaella Mistrulli, Francesco Antonio Veneziano, Francesco Grigioni, Massimo Volpe, Francesco Musumeci, Domenico Gabrielli

**Affiliations:** 1Department of Cardiosciences, A.O. San Camillo-Forlanini, 13449 Roma, Italy; fmusumeci@virgilio.it (F.M.); dgabrielli@scamilloforlanini.rm.it (D.G.); 2UniCamillus-Saint Camillus International University of Health Sciences, 00131 Roma, Italy; 3Cardiology Department, Sapienza University of Rome, Sant’Andrea Hospital, 00185 Roma, Italy; mistrulliraffaella@gmail.it (R.M.); massimo.volpe@uniroma1.it (M.V.); 4Unit of Cardiovascular Science, Department of Medicine, Campus Bio-Medico University, 00128 Roma, Italy; f.veneziano@unicampus.it (F.A.V.); f.grigioni@unicampus.it (F.G.)

**Keywords:** atrial fibrillation, acute coronary syndrome, dual antithrombotic therapy, triple antithrombotic therapy

## Abstract

Patients with atrial fibrillation (AF) are at increased risk for coronary artery disease (CAD). After percutaneous coronary intervention (PCI), the antithrombotic therapy consists of a combination of anticoagulant and antiplatelet agents to reduce the ischemic and thromboembolic risk, at the cost of increased bleeding events. In the past few years, several randomized clinical trials involving over 12,000 patients have been conducted to compare the safety of vitamin K antagonist (VKA) and direct-acting oral anticoagulants (DOACs) in association with a single- or double-antiplatelet agent, in the so-called dual- (DAT) or triple-antithrombotic therapy (TAT). These studies and several meta-analyses showed a consistent benefit for reducing bleeding events of DAT over TAT and of DOAC over VKA, without concerns about ischemic endpoints, except for a trend for increased stent thrombosis risk. The present paper examines current international guidelines’ recommendations and reviews clinical trials, meta-analyses, and observational studies conducted on AF patients treated with DAT or TAT after PCI for acute coronary syndromes.

## 1. Introduction

The incidence of atrial fibrillation (AF), the most common sustained arrhythmia, is increasing as a consequence of the aging population and wide-spreading risk factors [1,2]. AF increases the risk of thromboembolic events and major cardiovascular adverse events, and it is one of the most common causes of death globally and a leading cause of disability [3,4,5].

Up to 2/5 of patients with AF present a concomitant coronary artery disease (CAD). Moreover, about 15% of patients with AF may require percutaneous coronary interventions (PCIs) with stent implantation to treat obstructive CAD [6]. After PCI, these patients would require an antithrombotic therapy combining oral anticoagulant (OAC) and antiplatelet agents, aimed to decrease both the risk of thromboembolism due to AF and the risk of acute ischemic events related to thrombosis of coronary stents [7,8]. The benefits of this strategy may be counterbalanced by the increasing risk of bleeding [9].

In order to prevent stroke, OAC has proven to be more efficient and safer than aspirin monotherapy and dual-antiplatelet therapy (DAPT) [9,10,11,12]. Conversely, with regard to the reductions of ischemic events post PCI (i.e., recurrent ischemic events and stent thrombosis (ST)), DAPT is the treatment of choice [13,14,15,16]. Patients with AF on triple-antithrombotic therapy (TAT) experience higher rates of major bleeding (e.g., bleeding requiring hospitalization or fatal bleeding) compared with patients on dual-antithrombotic therapy (DAT) or single-antiplatelet therapy (SAPT) [8]. Therefore, the optimal strategy for this group of patients has to weigh the precarious balance of ischemic and haemorrhagic risk [17].

To date, six randomized controlled trials (RCTs), including a total of 11,421 patients, have been published to respond to this need [18]. The most promising approach to reduce bleeding events appears to be represented by the choice of a direct anticoagulant agent (DOAC) rather than vitamin K antagonists (VKAs) and to reduce the length of TAT or to avoid it.

## 2. Trials with VKAs

In the What is the Optimal antiplatElet and anticoagulant therapy in patients with oral anticoagulation and coronary StenTing (WOEST) study [19], an open-label, randomized trial, 573 patients (28% with ACS) receiving long-term therapy with warfarin (69% because of AF), with an INR target of 2.0, were randomly assigned to receive clopidogrel alone (*n* = 284) or clopidogrel and aspirin 80 mg/day (*n* = 289). The length of association treatment was one month after BMS placement and one year after DES placement (65% of patients). During the 12 mo of follow-up, the primary endpoint of any TIMI bleedings was less frequent in patients receiving DAT than TAT (hazards ratio (HR) 0.36, 95% CI 0.26–0.50, *p* < 0.0001), with no significant difference in TIMI major and GUSTO severe bleedings. Although it was not sufficiently powered for ischemic events, the rates of myocardial infarction (MI), stroke, target vessel revascularization, or ST did not significantly differ, but all-cause mortality at one year was lower in the dual-therapy group (2.5% vs. 6.4%; *p* = 0.027, HR: 0.39, 95% CI 0.16–0.93). Afterwards, in the Triple Therapy in Patients on Oral Anticoagulation After Drug Eluting Stent Implantation (ISAR TRIPLE) [20] trial, 614 patients (83% for AF or atrial flutter, 32% with ACS) were randomized to receive OAC for at least 12 mo and undergoing PCI with stent implantation (99% DES) were randomized to TAT with clopidogrel for 6 wk (*n* = 307) vs. 6 mo (*n* = 307), on top of aspirin and OAC, with the lowest recommended target international normalized ratio (INR) during the duration of TAT. At landmark analyses from 6 wk to 9 mo, the primary endpoint (death, MI, definite ST, stroke, or TIMI major bleeding) did not differ between groups (9.8% in the 6 wk group vs. 8.8% in the 6 mo group at 9 mo, HR: 1.14; 95% CI: 0.68 to 1.91; *p* = 0.63). No significant difference was registered in TIMI major bleeding (HR: 1.35; 95% CI: 0.64 to 2.84; *p* = 0.44), even though in the 6 wk group, there was a significantly lower risk of any Bleeding Academic Research Consortium (BARC) bleeding (HR: 0.68; 95% CI: 0.47 to 0.98; *p* = 0.04). As WOEST, also ISAR-TRIPLE has limited power to detect ischemic endpoints.

## 3. Trials with DOACs

The OPen-Label, Randomized, Controlled, Multicenter Study ExplorIng TwO TreatmeNt StratEgiEs of Rivaroxaban and a Dose-Adjusted Oral Vitamin K Antagonist Treatment Strategy in Subjects with Atrial Fibrillation who Undergo Percutaneous Coronary Intervention (PIONEER) AF-PCI [21] was the first open-label randomized trial testing DAT with DOACs vs. TAT with VKAs. In this trial, 2124 non-valvular AF patients undergoing PCI with stenting (66% DES, 40% with ACS) were randomized in a 1:1:1 ratio to receive: (1) rivaroxaban 15 mg daily + P2Y12 inhibitors or (2) 1 mo, 6 mo, or 12 mo of DAPT plus rivaroxaban 2.5 mg (twice daily, bid) followed by rivaroxaban 15 mg daily + aspirin, or (3) 1 mo, 6 mo, or 12 mo of DAPT plus VKA followed by VKA plus aspirin (full TAT arm, with INR target of 2.0-3.0). The TAT duration was decided and prespecified by treating clinicians (1 mo, 6 mo, or 12 mo). The P2Y12 inhibitor was mainly clopidogrel (93–96%) and was continued up to one year in 49% of patients in the full TAT arm. The primary endpoint of clinically significant bleeding at 12 mo was reduced by both rivaroxaban-based strategies compared to the TAT group (for Group 1, HR 0.59, 95% CI 0.47–0.76; *p* < 0.001, and for Group 2, HR 0.63, 95% CI 0.50–0.80; *p* < 0.001) driven by lower rates of bleeding requiring medical attention, but not TIMI major or minor bleedings. The rivaroxaban-based regimen resulted in a reduced risk of total bleeding events and recurrent hospitalization for adverse events. The rate of major adverse cardiovascular events (a composite of death from cardiovascular causes, MI, or stroke) was similar in the three groups, as rates for each component of the endpoint, but also in this trial, the power for detecting differences in ischemic events was low.

Afterwards, in The open-label, Randomized Evaluation of Dual Antithrombotic Therapy with Dabigatran vs. Triple Therapy with Warfarin in Patients with Nonvalvular Atrial Fibrillation Undergoing Percutaneous Coronary Intervention (RE-DUAL PCI) [22] trial, 2725 patients with non-valvular AF, treated with intracoronary stenting (66% DES, 40% ACS), were randomized to (1) DOAC with dabigatran 110 mg BID plus either clopidogrel or ticagrelor, (2) DOAC with dabigatran 150 mg BID plus either clopidogrel or ticagrelor, or (3) TAT with warfarin plus aspirin (≤100 mg daily) and either clopidogrel or ticagrelor. In the TAT group, aspirin was discontinued after 1 mo in the case of BMS and after 3 mo in the case of DES implantation. Ticagrelor was associated with dabigatran in 12% of patients. At 1 y, the incidence of the primary endpoint (first major or clinically relevant nonmajor bleeding event) was significantly lower in both dabigatran arms versus TAT group (HR: 0.52; 95% CI, 0.42 to 0.63; *p* < 0.001 for noninferiority; *p* < 0.001 for the superiority for the 110 mg DAT group and HR, 0.72; 95% CI, 0.58 to 0.88; *p* < 0.001 for the noninferiority for 150 mg the DAT group, respectively). The rates of major bleeding alone and total bleeding were significantly lower in both DAT groups than in the TAT group. There was no difference in the composite efficacy endpoint of thromboembolic events (MI, stroke, or systemic embolism), death, or unplanned revascularization among the DAT groups versus the TAT group (HR, 1.04; 95% CI, 0.84 to 1.29; *p* = 0.005 for noninferiority). Interestingly, when 110 mg BID dabigatran DAT was compared to TAT, there was a trend towards increased risk of MI (HR 1.51, 95% CI 0.94–2.41, *p* = 0.09) and definite ST (HR 1.86, 95% CI 0.79–4.40, *p* = 0.15); in contrast, there was no substantial difference between 150 mg BID dabigatran DAT vs. TAT.

The largest study conducted to date in AF patients undergoing PCI is the AUGUSTUS (an open-label, 2 × 2 factorial, randomized controlled trial to evaluate the safety of apixaban vs. vitamin K antagonist and aspirin vs. placebo in patients with atrial fibrillation and acute coronary syndrome and/or percutaneous coronary intervention) [23]. It was a two-by-two factorial randomized trial that randomized 4614 patients with AF to apixaban 5 mg BID versus VKAs (open-label) and aspirin versus placebo (blinded) on top of a P2Y12 inhibitor administered to all patients. Around 37% of patients presented with a PCI-treated ACS; 24% had medically managed ACS; 39% underwent elective PCI. All patients received a P2Y12 inhibitor, mainly clopidogrel (93% of cases). The median time from the index event to randomization was 6 d, while the TAT duration was 6 mo. At 6 mo, 10.5% receiving apixaban had an ISTH major or clinically relevant nonmajor bleeding event, as compared to 14.7% receiving VKA (HR 0.69; 95% CI, 0.58 to 0.81; *p* < 0.001 for noninferiority; *p* < 0.001 for superiority). Considering fatal and ischemic events, 23.5% who had been assigned to receive apixaban died or were hospitalized, as compared with 27.4% assigned to receive a VKA (HR 0.83; 95% CI, 0.74 to 0.93; *p* = 0.002). Although also this trial was not powered to assess efficacy, in PCI-treated patients, there was a trend toward a two-fold increased risk of probable or definite ST and toward an increased rate of MI in the DAT versus TAT arm ischemic event.

In the antiplatelet regimen comparison, the primary outcome of major or clinically relevant nonmajor bleeding was significantly increased by aspirin compared with placebo (HR: 1.89; 95% CI: 1.59 to 2.24; *p* < 0.001). Patients in the aspirin group had a 6 mo incidence of death or hospitalization similar to the placebo group (HR 1.08; 95% CI, 0.96 to 1.21). The incidence of death or hospitalization at 6 mo was higher among patients assigned to VKA and aspirin compared to those assigned to receive apixaban and placebo. Of note, nonsignificant 0.5% and 0.4% absolute increases in MI and ST, respectively, were noted in patients on placebo compared with those on aspirin. Therefore, AUGUSTUS helps to unravel the individual contribution of DOACs and aspirin withdrawal on the risk of bleeding. Importantly, compared with prior studies using DOACs, AUGUSTUS also included medically managed ACS patients, who are known to be at high ischemic risk. 

Finally, the Edoxaban Treatment Versus Vitamin K Antagonist in Patients With Atrial Fibrillation Undergoing Percutaneous Coronary Intervention (ENTRUST-AF PCI) study [24] was the last trial using DOACs in this setting: 1506 patients (52% with ACS) with AF undergoing PCI were assigned to edoxaban (60 mg once daily) plus a P2Y12 inhibitor (clopidogrel in 92% of cases) for 12 mo or a VKA in combination with a P2Y12 inhibitor and aspirin (100 mg once daily, for 1–12 mo, INR target 2.0-3.0). At 1 y, there were nonsignificant trends toward lower rates of the primary safety outcome (major or clinically relevant nonmajor bleeding) favouring DAT versus TAT (HR 0.83, 95% CI 0.65–1.05; *p* = 0·001 for noninferiority, margin hazards ratio 1·20; *p* = 0·11 for superiority). Although even this latter trial was not powered to assess ischemic outcomes, the main efficacy outcome (the composite of cardiovascular death, stroke, systemic embolic events, MI, and definite ST) was similar between the two groups (HR for edoxaban 1.06 (95% CI 0.71–1.69)). Interestingly, there was a very early numerical increase in ischaemic events in patients without aspirin, because of a nonsignificant trend toward increased risk of MI (HR 1.26, 95% CI 0.72–2.16) and ST (HR 1.32, 0.46–3.79) (Table 1).

## 4. Current Recommendations

### 4.1. European Guidelines

The 2020 European Society of Cardiology (ESC) guidelines on AF [4] recommend that in patients with AF and ACS undergoing an uncomplicated PCI, a TAT with aspirin, clopidogrel, and an OAC should be given during the peri-PCI period, up to 1 wk. Another option is to extend TAT up to ≤1 mo when the risk of ST outweighs the bleeding risk [25] (class of recommendation IIa, level of evidence C). Regardless of the initial treatment plan, a DAT with OAC and an antiplatelet drug (preferably clopidogrel) is recommended for the first 12 mo after PCI for ACS [26,27] or 6 mo after PCI in patients with CCS (class of recommendation I, level of evidence A). Thereafter, OAC monotherapy is to be continued (irrespective of the stent type) [20]. In the absence of contraindications, a DOAC should be preferred to VKA in combination with antiplatelet therapy (class of recommendation I, level of evidence A) [28]. In patients with a high bleeding risk (HAS BLEED ≥ 3), rivaroxaban 15 mg OD should be considered instead of rivaroxaban 20 mg OD or dabigatran 110 mg BID should be preferred to dabigatran 150 mg BID for the duration of concomitant antiplatelet therapy (class of recommendation IIa, level of evidence B) [21,29]. For patients with an indication for VKA in combination with antiplatelet therapy, the dose of VKA should be carefully monitored with an INR in the lower recommended range (2.0–2.5), and a good quality anticoagulation control, as reflected by high TTR (e.g., >70%). Notably, the use of ticagrelor or prasugrel as part of TAT is not recommended [30].

### 4.2. North American Guidelines

Updated guidelines and expert consensus documents in Europe and the United States are relatively aligned with respect to recommendations for antithrombotic therapy in patients with AF undergoing PCI [31]. The 2019 guidelines from American Heart Association/American College of Cardiology (AHA/ACC) suggest that patients with AF at increased risk of stroke (based on a CHA_2_-DS_2_-VASc risk score of two or greater) undergoing PCI with stenting normally require TAT. TAT should not go beyond a limited period of 4–6 wk, as this is the period of greatest risk of ST, especially in patients with ACS (class of recommendation IIb, level of evidence B-R). In this setting, it is recommended to choose clopidogrel instead of ticagrelor or prasugrel (class of recommendation IIa, level of evidence B-NR). An option is to use DAT with DOAC as an alternative to TAT to reduce bleeding [32,33] (class of recommendation IIa, level of evidence B-R). DAT is also recommended, instead of TAT, with VKA and clopidogrel or ticagrelor (class of recommendation IIa, level of evidence B-NR) [34,35].

### 4.3. North American Consensus on Antithrombotic Therapy in Patients with AF Undergoing PCI

In this consensus documents, the default strategy is to use OAC and DAPT in all patients with AF and who are treated with intracoronary stents, until hospital discharge [35]. After this period, it is recommended to withdraw aspirin, continuing DAT only. During the first month after PCI, there is an increased risk of thrombotic complications; therefore, it is reasonable to use TAT for one month in patients with high thrombotic risk and who have a reasonable low risk of bleeding. Anyway, it is not recommended to prolong TAT beyond one month after PCI [36,37]. Clopidogrel should be preferred as the P2Y12 inhibitor, but ticagrelor can be used in patients with high thrombotic risk and without high bleeding risk [38,39]. Accordingly, prasugrel should be avoided in patients concomitantly treated with an OAC. In patients who are eligible for a DOAC, this is preferred to VKA in combination with antiplatelet therapy. This recommendation stems from the consistent reduction in bleeding complications with a DOAC, including intracranial haemorrhage, without an apparent trade-off in efficacy [30]. The use of a lower dose is not recommended (with the exception of rivaroxaban 15 mg since this has been studied in randomized trials) [23]. The duration of DAT depends on the patient’s risk profile, and it may change from one year to six months after PCI in patients with high bleeding risk or low thrombotic risk. After this period, patients should discontinue antiplatelet therapy and receive OAT alone. Continuation of antiplatelet therapy as an adjunct to OAC beyond 1 y should be reserved only for selected patients with a high risk for ischemic recurrences and low bleeding risk (Figure 1) [40].

### 4.4. 2021. European Heart Rhythm Association Practical Guide

The recently published update of the European Heart Rhythm Association (EHRA) practical guide emphasises the concept that the choice of anticoagulant, as well as the duration of TAT, need to be personalized based on patient risk [41,42]. Notably, the assessment of bleeding risk should lead to correct reversible bleeding risk factors. Therefore, a careful follow-up can help to identify signs of occult bleeding. The treatment of patients undergoing an elective PCI is the same as for patients with ACS, but the transition to NOAC monotherapy should be made after six months instead of one year (Figure 2) [42].

## 5. Metanalyses

Following the publication of the dedicated randomized studies, fourteen metanalyses [24,36,38,43,44,45,46,47,48,49,50,51,52,53] have been conducted, but only four analysed all four RCTs with DOACs. In all the aforementioned studies, DAT, compared to TAT, conferred a significantly reduced risk of overall bleeding. Although two metanalyses [24,49] reported comparable cardiovascular events (all-cause death, stroke, ST, or MI) with DAT vs. TAT with DOACs, others found a significant increase in ST with the use of DAT. Notably, the two meta-analyses not reporting increased ST rates with DAT did not use the definition recommended by the Academic Research Consortium 2, resulting in debatable results. In one of the most recent analyses [36], the odds of ST with DAT vs. TAT increased 60%, with a number needed to harm of 274 and an associated borderline significant increase of MI. According to a subgroup pooled analysis [40] by the type of index event (ACS-related PCI vs. elective PCI), the risk of MI resulted in being higher in patients with AF and ACS undergoing PCI and receiving an early DAT, as compared to TAT [18].

## 6. Observational Studies

The Register of Information and Knowledge about Swedish Heart Intensive care Admissions (RIKS-HIA) [54] analysed 6182 patients with AF and ACS admitted to the coronary care units of 72 Swedish hospitals from 1995 to 2002. The majority of patients enrolled presented AF at admission (78%), while the remaining developed AF later (22%). Among patients with AF, 29% were prescribed a VKA alone or in association with aspirin (ASA), whereas 60% were given ASA and/or thienopyridine, and 11% did not receive any antiplatelet or anticoagulation therapy. Patients without any antithrombotic treatment had significantly higher 1 y mortality (45%) than patients receiving platelet inhibitors only (31%) or VKA (alone or in combination with platelet inhibitors; 22%). OAC treatment was associated with a 29% relative and a 7% absolute reduction in 1 y mortality, mainly caused by a reduction in fatal stroke.

Another study investigated patients with AF admitted with MI or PCI between 2000 and 2009, using data from nationwide registries in Denmark [55]. The main aim was to evaluate the risk and time frame of bleeding associated with TAT in comparison with other treatment regimens. The cohort was divided into five groups based on the treatment regimen: SAPT with aspirin or clopidogrel; VKA monotherapy; DAPT; DAT or TAT using VKA. This study showed a net increase of bleeding immediately after initiation of TAT compared to other regimens, whereas the rate of the combined end points of cardiovascular death, MI, and ischemic stroke was similar for TAT and DAT.

The Management of Antithrombotic TherApy in Patients with Chronic or DevelOping AtRial Fibrillation During Hospitalization for PCI ( MATADOR-PCI ) was a prospective, observational, nationwide registry of ACS treated with PCI and concomitant AF conducted in Italy between August 2018 and December 2019. Five hundred ninety-eight consecutive patients were enrolled: 292 (48.8%) with AF at hospital admission and 306 (51.2%) developing AF during the index hospitalization. Among this latter group, 131 (42.8%) developed AF before and 175 (57.2%) after PCI. Among the 211 patients with AF at admission, 116 (55.0%) had a permanent AF. DAPT, as the antithrombotic therapy at discharge, was prescribed in 26%, TAT in 65%, and DAT in 9% of patients. This latter strategy was mainly chosen for patients at high bleeding risk [56,57]. Clopidogrel was the most frequently used oral P2Y12 receptor inhibitor, while ticagrelor or prasugrel was prescribed in 2.3% and 13.5% of patients on TAT or DAT, respectively. DOACs were the most commonly used anticoagulant therapies both in patients receiving TAT or DAT. A low dosage of DOAC was employed in 71% of patients receiving TAT and 45% of those receiving DAT, but these prescriptions were appropriate only in 53% and 55% of patients receiving TAT or DAT, respectively [58]. Compared to the acute setting, the rate of TAT was significantly reduced (from 76.4% to 23.6% and from 53.8% to 23.6%; both *p* < 0.0001), while DAT was increased (from 11.8% to 56.3% and from 5.8% to 30.9%; both *p* < 0.0001) at 6 mo follow-up, in patients with pre-existing and new-onset AF, respectively [59].

## 7. Conclusions

TAT exposes AF patients to a high bleeding risk at 30 d follow-up. Several randomized clinical trials involving approximately 12,000 patients demonstrated that DAT significantly reduces bleeding events compared to TAT. Current evidence suggests that DOAC, at the dose recommended for thromboembolic protection in AF, should be preferred over VKA, while clopidogrel should be used when a TAT is prescribed. Further larger studies are needed to demonstrate if DAT might increase the risk of ST in certain subgroups of post-PCI patients. International guidelines and consensus documents recommend DAT as the default strategy in AF patients 1 wk after PCI, unless a high thrombotic and a low bleeding risk, needing TAT for a prolonged period of time, coexist. Observational studies suggest that TAT is still largely prescribed, while DAT is reserved for patients deemed at high bleeding risk. In this context, educational campaigns for the implementation of guideline recommendations are desirable.

## Figures and Tables

**Figure 1 jcm-11-00512-f001:**
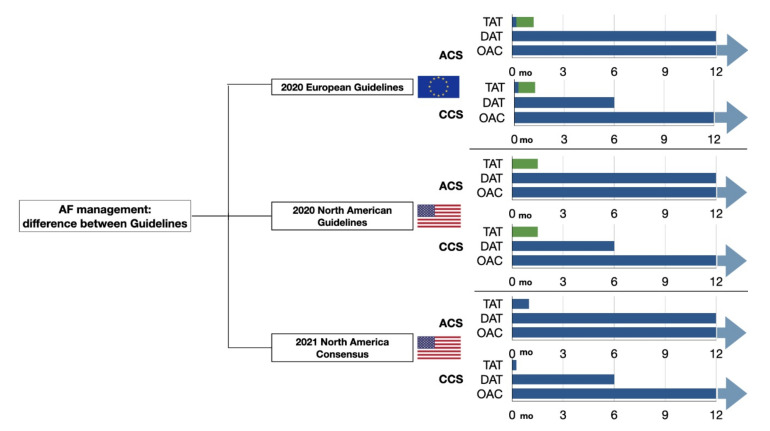
Antithrombotic therapy duration: differences across guidelines. ACS, acute coronary syndrome; AF, atrial fibrillation; CCS, chronic coronary syndrome; DAT, dual-antithrombotic therapy; OAC, oral anticoagulant; TAT, triple-antithrombotic therapy.

**Figure 2 jcm-11-00512-f002:**
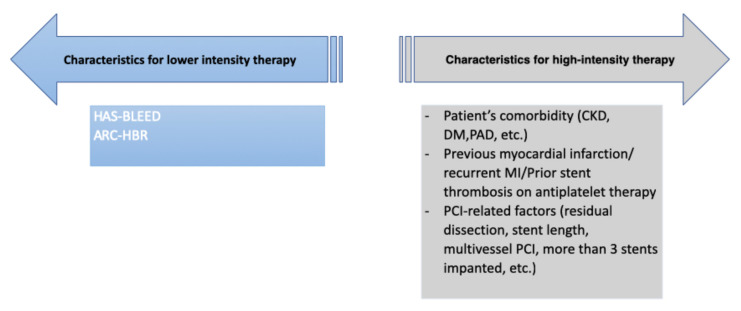
Risk factors associated with an increased risk of bleeding and an increased risk of ischemic coronary outcomes. ARB-HBR = The Academic Research Consortium for High Bleeding Risk; CKD = Chronic kidney disease; DM = Diabetes mellitus; PAD = Peripheral arterial disease; MI = Myocardial infarction; PCI = Percutaneous coronary intervention.

**Table 1 jcm-11-00512-t001:** Design of major trials comparing TAT vs. DAT.

	Design, Randomization	Patients (*n*)	Treatment Arms	AF (%)	Maximum Days in ASA in DAT Group	ACS (%)	Follow Up (mo)	Primary Outcome
WOEST	RCT open label, 1:1	573	VKA + P2Y12i for 1 or 12 mo vs. VKA + aspirin + P2Y12i for 1 or 12 mo	69	<1	28	12	Any bleeding events (TIMI and GUSTO criteria)
ISAR-TRIPLE	RCT open label, 1:1	614	6 we of VKA + clopidogrel + aspirin vs. 6 mo of VKA + clopidogrel + aspirin	84	/	32	9	A composite of death, MI, ST, stroke, or major bleeding
PIONEER-AF	RCT open label, 1:1:1	2124	rivaroxaban (15 mg/od) + P2Y12i vs. rivaroxaban (2.5 mg/bid) + aspirin + P2Y12i vs. VKA + aspirin + P2Y12i	100	3	52	12	Clinically significant bleeding (TIMI major, minor bleeding, or bleeding requiring medical attention)
RE-DUAL PCI	RCT open label, 1:1:1	2725	dabigatran (110 mg/bid) + P2Y12i vs. dabigatran (150 mg/bid) + P2Y12i vs. VKA + aspirin (1–3 months) + P2Y12i	100	5	51	14	Clinically significant bleeding (ISTH major bleeding or clinically relevant nonmajor bleeding event)
AUGUSTUS	RCT open label, 2 × 2	4614	apixaban (5 mg/bid) + P2Y12i vs. apixaban (5 mg/bid) + aspirin + P2Y12i vs. VKA + P2Y12i vs. VKA + aspirin + P2Y12	100	14	60	6	Clinically significant bleeding (ISTH major or clinically relevant nonmajor bleeding)
ENTRUST-AF PCI	RCT open label, 1:1	1506	Edoxaban (60 mg/od) + P2Y12i vs. VKA + aspirin (1–12 mo) + P2Y12	100	5	52	12	Major or clinically relevant nonmajor bleeding (according to ISTH criteria)

RCT = randomized controlled trial; MI = myocardial infarction; ST = stent thrombosis; ISTH = International Society on Thrombosis and Haemostasis; GUSTO = Global Utilization of Streptokinase and Tissue Plasminogen Activator for Occluded Coronary Arteries.

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
