# Peer review of "Antithrombotic Strategies in Patients with Atrial Fibrillation and Acute Coronary Syndromes Undergoing Percutaneous Coronary Intervention"

_jcm, 2022, doi:10.3390/jcm11030512_

Round 1

Reviewer 1 Report

De Luca and coworkers explore therapeutic strategies in the patient with atrial fibrillation undergoing coronary angioplasty. The manuscript is informative and interesting. Well written and with clear information to be transferred into clinical practice.

Author Response

Reviewer #1

De Luca and coworkers explore therapeutic strategies in the patient with atrial fibrillation undergoing coronary angioplasty. The manuscript is informative and interesting. Well written and with clear information to be transferred into clinical practice.

Authors reply: we thank the reviewer for the positive comments.

Reviewer 2 Report

While the review of Leonardo De Luca covers all major trials, meta-analyses and guideline recommendations of studies evaluating antithrombotic therapy in patients with an indication of OAC and PCI or ACS, there is much separation between the descriptions of the individual studies. I therefore recommend to adapt a review to highlight the similarities and differences between the studies and recommendations. Currently, there is no connection between the individual studies or recommendations. It would be also interesting to comment on the underlying evidence of individual recommendations. I furthermore recommend to include the EHRA Practical Guide on NOAC use by Steffel et al that rather favour 30-day triple therapy in contrast to ESC guidelines.

Author Response

Reviewer #2

While the review of Leonardo De Luca covers all major trials, meta-analyses and guideline recommendations of studies evaluating antithrombotic therapy in patients with an indication of OAC and PCI or ACS, there is much separation between the descriptions of the individual studies. I therefore recommend to adapt a review to highlight the similarities and differences between the studies and recommendations. Currently, there is no connection between the individual studies or recommendations. 

Authors reply: we thank the reviewer for the comment. we have now reviewed and modified the manuscript according to reviewer’s suggestions

I furthermore recommend to include the EHRA Practical Guide on NOAC use by Steffel et al that rather favour 30-day triple therapy in contrast to ESC guidelines.

Authors reply: we have now added a paragraph (page 10, last paragraph) with the EHRA practical guide recommendations.